# Boosting the Immune Response with the Combination of Electrochemotherapy and Immunotherapy: A New Weapon for Squamous Cell Carcinoma of the Head and Neck?

**DOI:** 10.3390/cancers12102781

**Published:** 2020-09-28

**Authors:** Francesco Longo, Francesco Perri, Francesco Caponigro, Giuseppina Della Vittoria Scarpati, Agostino Guida, Ettore Pavone, Corrado Aversa, Paolo Muto, Mario Giuliano, Franco Ionna, Raffaele Solla

**Affiliations:** 1Department of Otolaryngology Surgery and Oncology, Ospedale Casa Sollievo della Sofferenza, 71013 San Giovanni Rotondo, Italy; f.longo@istitutotumori.na.it; 2Head and Neck Medical Oncology Unit, INT IRCCS Fondazione G. Pascale, 80131 Naples, Italy; f.caponigro@istitutotumori.na.it; 3Medical Oncology Unit, Hospital of Pollena Trocchia, ASLNA3 sud, 80040 Naples, Italy; giuseppina.dellavittoria@gmail.com; 4Department of Otolaryngology Surgery and Oncology, INT IRCCS Fondazione G. Pascale, 80131 Naples, Italy; a.guida@istitutotumori.na.it (A.G.); e.pavone@istitutotumori.na.it (E.P.); c.aversa@istitutotumori.na.it (C.A.); f.ionna@istitutotumori.na.it (F.I.); 5Department of Radiation Oncology, INT IRCCS Fondazione G. Pascale, 80131 Naples, Italy; p.muto@istitutotumori.na.it; 6Department of Experimental and Clinical Oncology, University of Naples “Federico II”, 80131 Naples, Italy; m.giuliano@unina.it; 7Italian National Research Council, Institute of Biostructure & Bioimaging, 80131 Naples, Italy; raffaele.solla@ibb.cnr.it

**Keywords:** electrochemotherapy, immunotherapy, squamous cell carcinoma of the head and neck, abscopal effect, immunologic cell death

## Abstract

**Simple Summary:**

Squamous cell carcinoma of the head and neck (SCCHN) represents a problem of utmost concern and, for many clinicians and surgeons, an enormous challenge. Currently, new generation immunotherapy which avails of check point inhibitors, namely molecules capable of restoring the host’s immune system strongly depressed by the presence of tumor cells, is gaining increasing importance. Nevertheless, immunotherapy alone is not always effective in some patients, in particular those having a bulky and highly symptomatic disease. These last require the addition of locoregional strategies able to reduce the tumor mass and to assist immunotherapy in producing its effect. Electrochemotherapy (ECT) is a strategy able to associate the electroporation of tumor cells and the simultaneous administration of antineoplastic drugs, so as to concentrate the latter directly in the tumor site. The combination of ECT and immunotherapy could be very effective particularly in patients having a bulky/highly symptomatic SCCHN.

**Abstract:**

Head and neck squamous cell carcinomas (SCCHN) are not rare malignancies and account for 7% of all solid tumors. Prognosis of SCCHN patients strongly depends on tumor extension, site of onset, and genetics. Advanced disease (recurrent/metastatic) is associated with poor prognosis, with a median overall survival of 13 months. In these patients, immunotherapy may represent an interesting option of treatment, given the good results reached by check-point inhibitors in clinical practice. Nevertheless, only a minor number of patients with advanced disease respond to immunotherapy, and, disease progressions/hyper-progressions are common. The latter could be a very difficult issue, especially in patients having a wide and highly symptomatic head/neck mass. Given the potentiality to boost the immune response of some local modalities, such as electrochemotherapy, a possible future approach may take into account the combination of electrochemotherapy and immunotherapy to treat patients affected by SCCHN, suffering from symptomatic lesions that need rapid debulking.

## 1. Background

Squamous cell carcinoma of the head and neck (SCCHN) represents about 7% of all malignancies accounting for more than 650,000 cases and 330,000 deaths annually worldwide [1]. The main recognized risk factors are smoke, alcohol consumption, oral trauma, and Human Papilloma Virus (HPV) infection [2]. The therapeutic approach for SCCHN patients depends on disease location, staging (according to Tumor, Node, Metastasis system) and more frequently, by its genetics [3,4,5]. Despite multidisciplinary aggressive approaches including surgery and/or radiation therapy with or without chemotherapy/biological therapy/immunotherapy, the outcome of these patients is currently not satisfactory, considering that recurrences occur in about 20–50% of patients treated upfront, and in almost 95% of those who have already experienced a recurrent disease [6,7].

Immunotherapy is an effective therapeutic strategy, which has gained more and more importance over the last decade. The goal of immunotherapy is to reinforce the host immune system, leading it to react against tumor cells, thus provoking tumor elimination [8].

Several strategies of immunotherapy have been employed in past years, starting from the administration of soluble high-dose immune-stimulant cytokines, in the 70s and 80s, until the recent introduction of immune check-point inhibitors.

The rationale for the use of almost all immunotherapeutic strategies is the presence of the “so called” tumor associated antigens (TAA), which are tumor cell-produced proteins able to elicit an immune response [9], mainly mediated by cytotoxic-T lymphocytes.

Several strategies aimed to reinforce host immune response against cancer cells have been developed over the years, and all of them have common scope to generate strongly a class of T-lymphocytes (CD8+) selectively able to recognize TAA, and hence able to attack the tumor cells [10].

Immune response is not easy to be characterized, but it is now well acknowledged that within the whole process, there are two very important and crucial phases, named “check-points”. During these phases, once having recognized the TAA, the cytotoxic-T lymphocytes, become able to mature and attack the tumor cell. However, activated cytotoxic-T lymphocytes can be inhibited and pushed to anergy. In this way, the entire process arrests itself and the immune response is evaded.

The first check-point is the so called “priming phase” during which dendritic cells (DC) present the TAA to the naïve T-lymphocytes. During the priming phase, different tumor antigens are first internalized, then processed and finally presented by the DCs, through the class II major histocompatibility complex (MHC II), to the naive T lymphocytes. This signal alone is not sufficient to activate cytotoxic T-lymphocytes, but requires a second co-stimulatory signal, which is normally provided by the interaction of B.71 (present on the DC membrane) and CD28 (present on that of T-lymphocytes). This phase may be blocked when the immunosuppressive cytokines produced by the tumor cells or by the immunosuppressive lymphocytes (T-reg) recruited by the tumor cells, induce the expression of the Cytotoxic-T-Lymphocytes Associated antigen-4 (CTLA-4) rather than the costimulatory receptor CD28, on the cell membrane of cytotoxic T-lymphocytes. In turn, CTLA-4 interacts with the B7.1 on DC cell membrane inducing the cytotoxic T-lymphocytes anergy [11,12]. The second check-point corresponds to the so called “effector phase”, during which, the mature cytotoxic T-lymphocytes react directly with the TAA-exposing tumor cells, leading to their death by releasing granzyme, perforines or by linking the FAS-receptors expressed on the tumor cells surface, and ultimately inducing apoptosis [13,14]. During this phase, the T-lymphocytes, now mature and ready to interact with the tumor cells, are able to directly recognize the TAA presented to them directly by the tumor cells, through the MHC of class I. This unique interaction, which takes place between the TAA presented by the class I MHC and the T-Cell receptor (TCR) present on the membrane of the T-lymphocytes, is sufficient to activate the T-lymphocyte and cause the destruction of the tumor cell. The tumor cell, as an escape mechanism, can over-express PD-L1, which by interacting with the PD-1 present on the membrane of the T-lymphocyte, can cause its anergy.

Check-point inhibitors represent the new-generation immunotherapy for solid tumors. These drugs are capable of restoring the activity of the cytotoxic T-cells, acting in the check-point phases of the immune response, and leading to a re-activations of the above-mentioned immune cells. Despite positive results achieved by check-point inhibitors in clinical practice, several patients do not benefit from these therapies.

New immunotherapeutic combinations between different check-point inhibitors, such as the recognition of predictive factors of response to immunotherapy, should help clinicians in the near future, but the association of a locoregional therapy, able to boost the immune response against TAA, if associated with immunotherapy, may also represent an intriguing strategy.

## 2. Electrochemotherapy

Electrochemotherapy (ECT) is an anti-tumor strategy, which associates electroporation and concomitant delivering of antineoplastic drugs. Electroporation consists of the application of short-intensity pulsed electric fields to tumor cells, following which, the plasma membrane permeability to different hydrophilic drugs transiently increases, thus facilitating cellular uptake of cytotoxic agents [15,16]. The electric pulses, locally delivered to the whole tumor volume, are able to permeabilize tumor cells in a reversible manner. As a consequence, the anti-cancer drugs, administered directly into the tumor or systemically, can enter the electroporated target cells without restrictions, and can determine increased cell death [17]. The rationale which has prompted the use of ECT is based on the fact that tumor tissues have often an irregular vascularization, which causes an insufficient intake of antineoplastic drugs in the tumor site. Other mechanisms of action are the “vascular lock” causing a strong vascular spasm, which leads to the interruption of tumor bleeding and to a prolonged contact time between the drug and the tumor tissues [18], and finally “immunologic cell death”. Indeed, ECT is able to activate immune responses against different TAA indirectly. Tumor cell destruction, during ECT, leads to the exposure of several immunogenic antigens, which can recruit from peripheral blood antigen-presenting cells (APC) and DC, eliciting a robust immune response against the tumor [19,20]. This mechanism provides the rationale for the combination of ECT and immunotherapy (Figure 1).

## 3. Boosting the Immune Response against Cancer: Biological Mechanisms and Data from the Literature

ECT is highly efficient in provoking tumor shrinkage, nevertheless it remains a local treatment having no apparent anti-tumor effects on non-treated lesions. Interestingly, there is compelling evidence that the immune system crucially contributes to ECT efficiency. In fact, preclinical studies highlighted that ECT-mediated tumor regression was dramatically decreased in animals depleted of functional T lymphocytes, in comparison to immunocompetent mice [21,22,23,24]. In addition, several authors observed a significantly wider edema and inflammatory reaction elicited by the ECT, in immune-competent compared to immune-deficient mice. More interestingly, a strong peripheral release of circulating monocytes was shown in models of murine fibrosarcoma, after ECT [25,26,27].

These data have suggested the hypothesis that ECT is able to strongly activate the immune system after its administration. The potential mechanisms responsible for this phenomenon have been in part elucidated and have taken into account the capability of ECT to provoke “immunogenic cell death” (ICD). Different from normal apoptosis, that is mostly tolerogenic or non-immunogenic, ICD is characterized by the tissue expression of different immunogenic antigens and a strong recruitment of DC and APC. ICD is characterized by tumor cell secretion of damage-associated molecular patterns (DAMPs), which are proteins activated in response to particular stress, such as electroporation. These proteins, including calreticulin, heat shock protein 70 (HSP70), and HSP90, once exposed on the tumor cell surface, exert an immune-stimulatory effect, based on their interaction with APC. In detail, electric pulses delivered to tissues create an inflammatory microenvironment that could facilitate the infiltration of macrophages, polymorphonuclear leukocytes, and specifically, through the release of ATP by the electroporated cells, DC and APC. Once recruited, these cells, which are also stimulated by DAMPs, process and present the TAA to cytotoxic t-Lymphocytes.

As an additional mechanism, it has been hypothesized that also immune-stimulant cytokines, such as IFN-Gamma, IL-2 and TNF-Alpha, released by the electroporated tumor cells, are able to both recruit and activate the APC [28,29,30,31,32].

Briefly, ECT causes the release of TAA and their presentation by the APC to the T-lymphocytes, thereby stimulating an immune response. However, this immune response does not appear to be robust enough to cause regression of untreated distant tumor lesions, probably due to the absence of memory cytotoxic T lymphocytes, not sufficient immune-stimulant cytokine upregulation or the low number of specific cytotoxic t-lymphocytes produced. This consideration has led to studies investigating the use of ECT in combination with immunotherapy, first in animal models and more recently in human patients, with the aim to determine whether this therapeutic combination could increase the ECT-induced immunological response and lead to a systemic antitumor response (Figure 2).

First, some studies have investigated the role of ECT combined with immunotherapy as a novel treatment strategy for metastatic melanoma [33].

Mir et al. [34] examined the combination of ECT and soluble IL-2, in comparison with ECT alone, in mice bearing fibrosarcoma. As a result, the therapeutic combination led to both a local and a systemic response, as evidenced by regression of distant nodules not treated by ECT and the presence of significant CD4+ and CD8+ T-cell infiltrates in non-treated nodules.

Brizio et al. [35] described a case of a patient with advanced melanoma and multiple cutaneous metastases who experienced a complete clinical response to the combination of ECT plus ipilimumab.

Mozzillo et al. [36] conducted a retrospective analysis of patients treated with both ipilimumab and ECT. Fifteen patients with previously treated metastatic melanoma who received ipilimumab 3 mg/kg every three weeks for four cycles and underwent ECT for local disease control and/or palliation of cutaneous lesions with bleomycin 15 mg/m^2^ after the first ipilimumab infusion, were included in the analysis. A local objective response was observed in 67% of patients (27% complete response (CR) and 40% partial response (PR)). The authors concluded that the combination of ipilimumab and ECT appeared to be beneficial in patients with advanced melanoma, as this novel treatment obtained a fairly good response rate and DCR. The only limitation of this study was the absence of a control arm containing ipilimumab alone.

Karaca et al. [37] published a case report of a patient with metastatic melanoma who received ECT plus a PD-1 inhibitor (nivolumab) as a fifth-line treatment. Remarkably, despite the patient having been previously treated with several lines of chemotherapy, targeted therapy, and immunotherapy, he achieved a complete response with no evidence of cutaneous or visceral disease at a 4-year follow-up period.

Heppt et al. [38] evaluated the use of ECT with a CTLA-4 inhibitor (ipilimumab) versus ECT with PD-1 inhibitors (pembrolizumab or nivolumab) in the treatment of unresectable or metastatic melanoma, in a retrospective trial. The ipilimumab cohort demonstrated a systemic overall response rate (ORR) of 19.2%, whereas the anti-PD-1 cohort showed a systemic ORR of 40%. The authors concluded that the association of ECT plus PD-1 inhibitors was more effective than ECT plus ipilimumab in terms of objective responses.

Unfortunately, we have only results carried out by case reports and retrospective trials (Table 1), but currently, a number of prospective clinical trials coupling inhibitors of the PD-1/PDL-1 axis and ECT are ongoing upon patients affected by several solid tumors, and, once published, we will have a better knowledge regarding the potential of the combination.

## 4. The Landscape of Immuno-ECT in SCCHN

ECT is currently employed in several solid tumors, including melanoma, basal and squamous cell carcinoma, Kaposi’s sarcoma, breast cancer, and, overall in cutaneous metastases. Its role is mainly palliative in cases of bleeding and painful masses. ECT can be applied to mucosal head and neck recurrent tumors accessible to the procedure using particular electrodes able to easily reach the head and neck anatomical regions. The most employed chemotherapeutic drug in clinical practice is bleomycin, and it is injected intravenously 8 minutes prior to the electric pulsed administration, according to the European standard operating procedures for the electrochemotherapy (ESOPE) guidelines [39,40]. Small case series of electroporation combined with bleomycin therapy in head and neck cancer have been reported in the literature with very promising results [41,42,43]. In particular, in a prospective trial of six European institutions [44], ECT was investigated in 36 patients with recurrent and mucosal head and neck cancers, most of them being primitive squamous cell skin cancers. An ORR of 56% was observed with a CR rate of 19%, a PR rate of 37%, and an SD rate of 23%. Three patients (7%) maintained their CR at 30, 34, and 84 months post-treatment, respectively.

Interestingly, Longo et al. [45], used bleomycin-based ECT to treat 93 patients with recurrent and/or metastatic head and neck tumors, mostly constituted by SCCHN, who had progressed after at least two lines of chemotherapy. Primary endpoints were palliation of the symptoms (bleeding and pain) and improvement of quality of life; secondary endpoints were ORR and DCR. A good control of pain and bleeding was obtained, especially in patients with moderate symptoms before the treatment and no toxicities related to ECT were seen. More interestingly, the CR rate was 5% and the PR rate was 40%, leading to a promising ORR rate of 45%. The DCR rate was also remarkable (79%). The authors concluded that ECT was particularly effective in palliating the symptoms and ameliorating the quality of life (QoL), and it was shown to be a very active treatment, in heavily pre-treated patients. Although retrospective, this study was one of the few clinical trials assessing ECT activity in SCCHN.

Unfortunately, there are, currently, no data assessing the activity and the efficacy of ECT plus immunotherapy in SCCHN, since immunotherapy has only recently been approved for SCCHN treatment. Therefore, phase II and III clinical trials assessing the combination of ECT and immunotherapy in SCCHN are strongly warranted.

## 5. Discussions and Conclusions

Immunotherapy has gained ever increasing importance in the clinic scenario of all solid tumors, including SCCHN. “New generation” immunotherapy, which mainly acts on the tumor microenvironment (TME), causing an immune-stimulation, is based on check-point inhibitors use. Nivolumab and pembrolizumab are currently the only approved drugs for SCCHN, as they both have been demonstrated to significantly prolong survival in patients with advanced disease [46,47]. Despite the good results obtained in this poor prognosis category of patients, check-point inhibitors do not always function and in some cases disease progression and hyper-progressions are observed [48]. Moreover, the response rate reached a maximum value of 13–14% in the Keynote 141 trial [46]. This latter is considered to have been a seminal and very important trial in head and neck oncology. It enrolled patients with recurrent/metastatic SCCHN, who had progressed within 6 months of a first-line therapy containing platinum. Patients were randomized to receive nivolumab or in alternative second line chemotherapy chosen by the experimenter. As results, nivolumab significantly prolonged OS and PFS, and it also significantly ameliorated ORR, if compared with the standard second-line chemotherapy.

Patients with recurrent SCCHN localized to the head/neck region may be particularly symptomatic and the most common symptoms are bleeding and pain. In addition, the head/neck region is rich in vascular and nervous vital structures, such as the carotid artery and the Vagus nerve, respectively, which could at an early stage be compressed and/or infiltrated by the tumor mass, thus provoking serious clinical complications. Based on these findings, in the presence of a wide and symptomatic head/neck mass, rapid debulking is mandatory, and immunotherapy alone is not able to guarantee it.

Several studies have tested the combination of immunotherapy and local symptomatic treatments in different solid tumors (Table 2). Theurich et al. [49] retrospectively analyzed clinical data from 127 consecutively treated melanoma patients at four cancer centers, who received either ipilimumab or ipilimumab with additional local treatment (stereotaxic radiotherapy or ECT). As results, the addition of local treatments to ipilimumab significantly prolonged overall survival (OS 93 vs. 42 weeks, *p* = 0.0028). The conclusions were that the addition of local treatments to ipilimumab was safe and effective in patients with advanced disease suffering from symptomatic masses, and importantly, the combined strategy was able to obtain a significantly higher response rate.

In addition to the capability of maximizing the volume reduction of symptomatic lesions, local treatments can also boost the immunogenicity of the tumor, by increasing the release of TAA, and thus stimulating a systemic response against distant nodules. This phenomenon, known as the “abscopal” effect, was described for the first time in association with radiation therapy [56], but the same mechanism is valid also for ECT. In fact, both radiotherapy and ECT are able to induce ICD and thus to boost the immune response against cancer. Most of the available data on the efficacy of combining ECT plus immunotherapy have been generated so far in patients affected by melanoma, as the immunotherapy has been employed in clinical practice for a long time for this disease.

In contrast, few data are available in SCCHN patients who may benefit from this new treatment strategy. In fact, SCCHN, and particularly those related to alcohol and tobacco consumption, are characterized by a sharp local immunosuppression, since the tumor tissues are often not infiltrated by cytotoxic T-lymphocytes, or alternatively, they are infiltrated by immunosuppressive lymphocytes, namely the T-Reg [57,58]. Therefore, these tumors may have a better response to immune therapies in the presence of a local treatment able to inflame the tumor and thus to boost the immune response.

The results of the Keynote 048 Study have been published recently. The study design included the comparison between pembrolizumab alone or its combination with chemotherapy versus standard chemotherapy (EXTREME scheme) as first-line treatment in patients with recurrent/metastatic SCCHN.

The authors discovered that pembrolizumab alone achieved a significantly higher OS if compared with standard chemotherapy (14.9 vs. 10.7 months, *p* < 0.0007) in patients whose tumor over-expresses tissue PD-L1 (Combined Positive Score >20). Moreover, the combination of pembrolizumab and chemotherapy was better than standard chemotherapy (13.0 vs. 10.9 months, *p* < 0.03) in the Intent to Treat (ITT) population (independently of the PD-L1 expression). These results paved the way for the recognition of chemo-immunotherapy as a potential new-standard first line therapy in recurrent/metastatic SCCHN. Nevertheless, the response rate obtained by immunotherapy alone and immune-chemotherapy was not satisfactory and neither pembrolizumab alone, nor pembrolizumab plus chemotherapy were superior to standard chemotherapy from this point of view [59].

We believe that an intriguing step forward may be to test the combination of ECT with new-generation immunotherapy (check-point inhibitors), especially in patients affected by SCCHN and wide symptomatic masses.

In the near future, in addition to the ECT-immunotherapy combination, other ways of manipulating the response to immunotherapy could be employed, and some of them are particularly simple and safe.

The gut microbiota deserves a special mention. In fact, evidence is growing that the gut microbiota can modulate the host response to cancer immunotherapy [60]. Gut flora or gut microbiota are the microorganisms including bacteria, archaea, and fungi that live in the digestive tracts of humans. These microorganisms have been shown to interact with one another and with the host immune system in ways that influence the development of disease. Moreover, recent data point out that patients with specific intestinal microflora have better responses to immunotherapy [61,62] and some authors have tried to identify microbes associated with immunotherapy responsiveness [63]. Techniques capable of modifying the gut microbiome and at the same time making the host more responsive to immunotherapy, are being developed.

## Figures and Tables

**Figure 1 cancers-12-02781-f001:**
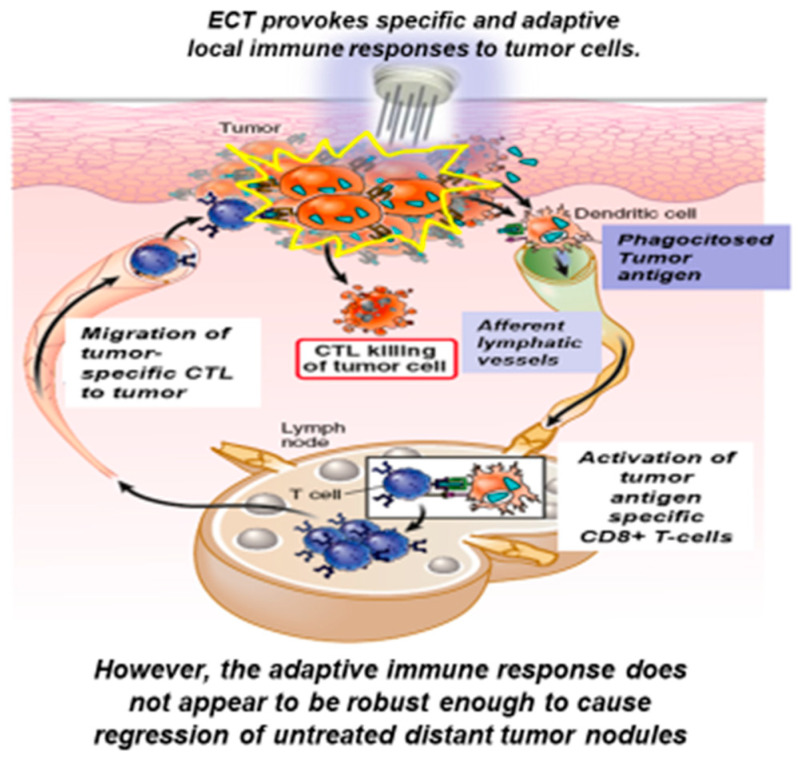
Description of the immune-stimulant effect exerted by electrochemotherapy.

**Figure 2 cancers-12-02781-f002:**
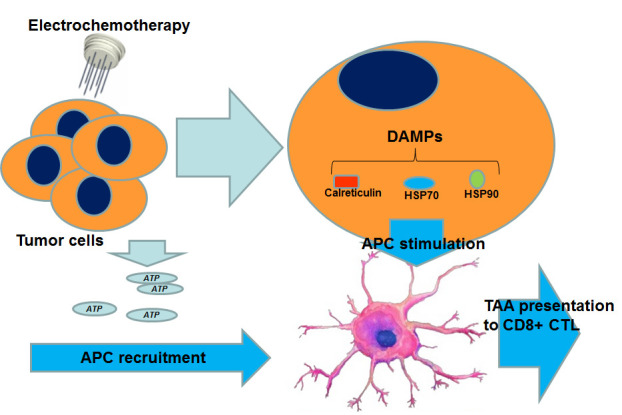
Mechanisms through which ECT may boost the immune response. APC can be stimulated directly through the release of ATP by tumor cells and also indirectly, through the intra-cellular production of DAMPs. (ATP: adenosine triphosphate; DAMPs: damage associated molecular pathways; APC: antigens presenting cell; CTL: cytotoxic T-lymphocytes).

**Table 1 cancers-12-02781-t001:** Literature findings concerning studies of electrochemotherapy (ECT) + immunotherapy.

Published Paper	Target	Treatment	Type of Study	Number of Patients	Responses (ORR)
Oncoimmunology 2015; 4: e1008842. Mozzillo et al. [36]	Advanced Melanoma	ECT + ipilimumab	Retrospective trial	15	67%
Anticancer Drugs 2017; 29: 190–196. Karaca et al. [20]	Advanced Melanoma	ECT + nivolumab	Case report	1	Not Applicable
Eur J Dermatol 2015; 25: 271–272. Brizio et al. [35]	Advanced Melanoma	ECT + ipilimumab	Case report	1	Not Applicable
Cancer Immunol Immunother 2016; 65: 951–959. Heppt et al. [38]	Advanced Melanoma	ECT + ipilimumab vs. ECT + anti-PD-1	Retrospective trial	33	59.2% adding the values obtained in the two arms

**Table 2 cancers-12-02781-t002:** Studies enrolling patients with head and neck carcinomas (included primitive skin cancers).

Published Papers	Target	Treatment	Type of Study	Number of Patients	Responses (ORR)
Acta Derm Venereol	Skin cancer	ECT	Retrospective trial	33	100%
2019. 1; 99(13): 1246–1252.					
Bonadies et al. [50]					
Head Neck 2019; 41(2): 329–339.	SCCHN	ECT	Phase II	26	58%
Plaschke et al. [51]					
Eur J Cancer 2017 Dec; 87: 172–181.	SCCHN	ECT	Phase II	36	56%
Plaschke et al. [52]					
J Transl Med 2017 Apr 26;15(1):82.	Skin cancer	ECT	Retrospective trial	22	81.8%
Di Monta et al. [53]					
Ann Surg 2012 Jun;255(6):1158–64.	Skin cancer	ECT	Phase II trial	25	100%
Gargiulo et al. [54]					
Oral Oncol. 2019 May; 92: 77–84.	SCCHN and Skin cancer	ECT	Retrospective trial	93	45%
Longo et al. [45]					
Dermatol Surg. 2010 Aug; 36(8): 1245–50.	Skin cancer	ECT	Retrospective trial	6	83%
Landstrom et al. [55]					

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
