# Peer review of "Boosting the Immune Response with the Combination of Electrochemotherapy and Immunotherapy: A New Weapon for Squamous Cell Carcinoma of the Head and Neck?"

_cancers, 2020, doi:10.3390/cancers12102781_

Round 1

Reviewer 1 Report

This review aims to provide and summarize the rationale and clinical evidences, respectively, for the use of electrochemotherapy in combination with immunotherapy such as checkpoint inhibitors in head and head neck cancer. The topic is highly clinically relevant since novel strategies are needed to boost the unlocked host immune response by checkpoint inhibitors.  The manuscript is almost well written and clear. Appropriate references are summarized and listed in the manuscript. However there is the need to carefully revise the manuscript since many inaccuracies are present. Only some of them are listed. Lastly a better description of mechanistic immune processes is also needed. 

Minor Comments

  • Page 1, line 11,12,13,14, 15: the authors should decide whether indicate city and State of the institutions were they come from.
  • Page 1, line 8: the corresponding author should indicate the international prefix of the phone number.
  • Page 1 line 22: the sentence “Advanced disease (recurrent/metastatic) is associated to poor prognosis, with a median overall survival of 13 months in clinical trials” should be read Advanced disease (recurrent/metastatic) is associated to poor prognosis, with a median overall survival of 13 months”.
  • Page 1 line 26: The sentence “respond to immunotherapy, and, disease progressions and/or hyper-progressions” should be read “respond to immunotherapy and disease progressions/hyper-progressions”
  • Page 1 line 36: represent should be read represents
  • Page 1 line 36: about 6%. This data is contrasting with 7% reported in the abstract.
  • Page 1 line 40: Tumor, Node, Metastases system should be read Tumor, Node, Metastasis system
  • Page 1 line 40: by it’s genetics should be read by its genetic.
  • Page 1 line 41: the term containing does not appear to be appropriate. I would suggest including.
  • Page 2 line 43: the sentence occur in about 20-50% of patients treated upfront, and in should be occur in about 20-50% of patients treated upfront and in
  • Page 2 line 47: the term tumor rejection does not read well
  • Page 2 line 51: The sentence The rationale for the use of of all immunotherapy should be read The rationale for the use of all immunotherapeutic strategies. Furthermore I would suggest to no state a so much broad conclusion about the presence of TTA for all immunotherapeutic strategies.
  • Page 2 line 52: tumor cells-produced proteins should be read tumor cell-produced proteins.
  • Page 2 line 54: immune response against cancer should be read host immune response against cancer cells.
  • Page 2 line 63: T-Lymphocytes should be read T-lymphocytes. This comment applies to several parts of the manuscript.
  • Page 2 line 66: antigen-4 () rather should be readantigen-4 (CTLA-4) rather
  • Page 2 line 67: interact should be read interacts.
  • References 11 and 12 are not appropriate.
  • Page 2 line 77: New immunotherapy associations shoul be read New immunotherapeutic associations
  • Page 2 line 78: factors predictive should be read predictive factors
  • Page 2 line 84 the plasma membranes permeability should be read the plasma membrane permeability.
  • Page 2 line 90: which causes the insufficient should be read which causes an insufficient
  • Page 3 line 94: ECT, is able should be read ECT is able. Tumor cells destruction shoul be read Tumor cell destruction
  • Page 2 line 95: the sentence to several immunogenic antigen exposure does not read well.
  • Page 2 line 97: The mechanism should be read This mechanism.
  • Page 3 line 115: take should be red taken.
  • Page 3 line 116: Abbreviation should be used when the word is mentioned for the first time. This comment applies to many words present in the manuscript such as overall survival (OS), immunogenic cell death (ICD), etc.
  • Page 4 line 122: edema should be replaced with an inflammatory microenvironment.     
  • Page 4 lie 140: with with should e read with
  • Page 5 line 183: It is not clear why there are multiple references in the sentence In particular, in a prospective trial of six European institutions 40-43, whether the trial is just one.
  • Page 5 line 186: the sentence CR at 30, 34, and 84 should be read CR at 30, 34 and 84
  • Page 6 line 192: the sentence More interestingly, of CR rate was 5%, and PR rate was 40%, should be read More interestingly, CR rate was 5% and PR rate was 40%,
  • Page 6 line 193: ORR rate of a 45% should be read ORR rate of 45%    
  • Page 6 line 194: QoL should be spelt when it is describe for the first time (see line 190).
  • Page 6 line 195: the sentence needs to be rewritten .
  • Page 6 line 196: the sentence activity also in the SCCHN should be read activity in the SCCHN.  
  • Page 6 line 209: reference 6 does not described the Keynote 141 trial. It should be replaced by more appoprite reference.
  • Page 6 line 236: namely the T-Reg (T-regulatory) should be read namely the T-Reg.
  • Page 7 line 243: the sentence The authors discovered that pembrolizumab alone was achieved a significantly does not read well.
  • PDL-1 should be read PD-L1
  • Line 250: nor should be read or.

Major comments

  1. The authors described the mechanism of host immune response making a point of the inhibition of the checkpoint molecules to enhance the immune response. First no mention on the role of HLA class I and II is described in processing and presenting TAA or better epitopes of TAA in the context of HLA class II for DC and HLA class I for cancer cells. In addition a description of the function of CTLA-4 in the priming phase is mentioned as a checkpoint molecule. However no description of chechpoint molecules such as PD-1/PD-L1 axis is present during the description of effector phase. This part should be described since later in the mansucrpt authors described the role of anti-PD-1/PD-L1 therapy in head and neck cancer. The terms PD-1 and PD-L1 should be also spelt.
  2. In Figure 1: Tumor cell presence is able to elicit A double immune response should be read Tumor cell presence is able to elicit a double immune response. In addition is not clear what does it mean ECT does provoke the same effect upon tumor cells. The same effect of what? Lastly However this immune response…… the sentence cannot be applied to all figure since adaptive and innate response elicit a systemic host immune response.
  3. Recruitment of DC by ICD is caused by both expression of TAA and release of DAMPs o cytokines. This part needs to be better described.

Author Response

Dear Editor and dear Reviewer,

we appreciated the suggestions for improving the manuscript, and therefore made all the required revisions. In particular we have:

1)added all the minor concerns, as suggested by the reviewer

2) added a section examining the role of HLA (I and II) and importantly we have also added a section highlighting the role of PD-1-PDL-1 axis.

3) we have modified both the figures and the figure legends, as suggested

we thank you so much for your help

Best Regards

Dr Francesco Perri (corresponding author)

Reviewer 2 Report

Dear Authors, 

The paper submitted is very interesting, it is well written and logically constructed. The aim of the manuscript is clear and the role of immunotherapy together with combination with electrochemotherapy is very interesting and fascinating as well. However, I have some suggestion as minor revision to improve and reinforce the background of your "short communication".

Indeed, some reference in the "background" section should be added.

For example, the sentence at lines 45-47 page 2, concerning immunotherapy in cancer need some references, such as Ward et al., 2014 "Clinical experience of cancer immunotherapy integrated with Oleic Acid complexed with de-glycosylated vitamin D binding protein"(doi:10.3844/ajisp.2014.23.32). 

Furthermore, the sentence at lines 54-56 page 2, need a reference like the one by Jackson et al., 2014 "Targeting CD8+ T-cell tolerance for cancer immunotherapy" doi: 10.2217/imt.14.51.

Figure 1 is fine, but is very similar to the one previously published (Perri et al., 2020 "Immune Response Against Head and Neck Cancer: Biological Mechanisms and Implication on Therapy" doi: 10.1016/j.tranon.2019.11.008). Please, modify it or quote the paper.

Finally, please fix the typos present in the paper such as adding the abbreviation in the brackets at line 66, page 2.  

Author Response

Dear Editor, dear Reviewer,

we appreciated the suggestions for improving the manuscript, and therefore made all the required revisions. In particular we have:

  • Added more citations, and in perticular we have added the reference number 8 as you requested

  • Added the reference number 10, to enrich the manuscript

  • Cited the paper (Perri et al) in the figure 1 legend

  • Performed the requested correction

we thank you so much for your help

Best Regards

Dr Francesco Perri (corresponding author)

Reviewer 3 Report

This manuscript presents evidence that phase II and III clinical trials assessing the combination of ECT and checkpoint blockade immunotherapy in SCCHN are warranted.  (If no phase I trials are completed, phase II and III will have to wait.)

Please have a native English speaker or professional English editing service revise this manuscript.  

"the" is overused

"association" should be replaced throughout the manuscript with "combination"

There are at least 20 incidences of improper wording and faulty punctuation in the first 40 lines alone.  In lines 43-93, there are at least 78, and so on...

Figure 1 has two images that are redundant--simply use the second image, and change the title to "ECT provokes specific and adaptive local immune responses to tumor cells."

If tumor-fighting immune cell recruitment to tumors is a weakness of ECT (has this aspect been directly studied?), please state so--otherwise, let the readers know that ECT effects on immune cell recruitment is unknown.  Figure 1 includes "Migration of tumor-specific CTL to tumor"--if this effect has been shown, could the authors speculate why CTLs do not find metastases (no memory CTLs or appropriate cytokine upregulation, perhaps?).

Citations should be carefully checked to be sure that each citation supports claims made by the authors.  The use of citations 19-25 lends strength to the authors' proposal of combining ECT with checkpoint blockade immunotherapy (CBI) therapy, but the citations do not align directly with the statements made in lines 105-112. For example, "In fact, preclinical studies have highlighted that ECT-mediated tumor regression was dramatically decreased in animals depleted of functional T lymphocytes, in comparison to immunocompetent mice." (lines 107-109) is really only supported by citation #20, not #s 19-22.  Another example--"More interestingly, a strong peripheral release of circulating monocytes was shown in models of murine fibrosarcoma, after ECT." (authors' citations here are #s 23-25).  Citation #23 is a dendritic cell/Langerhans cell recruitment clinical (human, not mouse) study of melanoma, not fibrosarcoma.  Citation #24 (perhaps the only citation that should have been used here) is a study of ECT with CpG injection in mouse models of melanoma and sarcoma, and citation #25 is a study of murine colon cancer cells.  

Please state the results from the Mozzillo study (citation #34) more clearly for the readers (lines 146-156):  the treatment groups and resulting CRs and PRs are confusing as written.

Please add results in a column (CRs, PRs, etc.) to Tables 1 and 2.

Please describe the Keynote 141 trial (line 209, # patients, treatments, etc.) for the readers.

Lines 249-252 make no sense to this reviewer--please re-word.

A number of prominent published reports showing the importance of host flora for successful checkpoint blockade immunotherapy suggest that this manuscript should include a statement acknowledging that future studies of ECT combined with CBI should collect data on gut microbiota and associated metabolic effects (on ATP release from tumor cells, for example?).

Author Response

Dear Editor, dear Reviewer,

we appreciated the suggestions for improving the manuscript, and therefore made all the required revisions. In particular we have:

  • we delivered the manuscript to a native speaker who revised it and corrected any imperfections
  • we have modified the Figure 1 according to the reviewer’s suggestion
  • we have, according to the reviewer’s comments, explained in the text the reason for which ECT does not provoke distant lesions disappearance
  • we have checked every citation, and, where the title of the bibliographic note does not seem to recall the text, we point out that within the quoted work there is text that justifies the quotation
  • we have checked the results of the Mozzillo’s study and we have corrected the data
  • we have added the results concerning the ORR in a separate column, in both the tables
  • we have clearly described in the text the Keynote 141 study
  • we have added in the “conclusions” paragraph a section explaining the impact of gut microbiome in this category of patients

we thank you so much for your help

Best Regards

Dr Francesco Perri (corresponding author)

This manuscript is a resubmission of an earlier submission. The following is a list of the peer review reports and author responses from that submission.